# Conflict, caregiver violence and gendered parenting: A cross-sectional study among adolescent girls and young women participating in a girls' empowerment programme in Myanmar

Isabelle Pearson[1]*, Elaine Chase[2], Cing Van Kim[3], Nang Ma San[3], Hkawn Ja[3], Ei Ei Soe[3], Khin Lae[3], Nandar Oo[3], Zin Mar Hlaing[3], Brooke Zobrist[3], Cathy Zimmerman[1], Meghna Ranganathan[1]

1 Department of Global Health and Development, London School of Hygiene and Tropical Medicine, London, United Kingdom, 2 Institute of Education, University College London, London, United Kingdom, 3 Girl Determined, Yangon, Myanmar

☯ Cathy Zimmerman and Meghna Ranganathan share last authorship
* Isabelle.Pearson@lshtm.ac.uk

## Abstract

Across Myanmar, armed conflict and political instability have exacerbated poverty, food insecurity, and disrupted essential social protection services, severely affected people's health and wellbeing. This is especially pertinent for girls and young women, for whom gender inequalities are exacerbated during conflict, increasing their risk of various forms of violence. We aimed to measure the prevalence of parent/caregiver-perpetrated violence against adolescent girls across Myanmar and examine its association with exposure to conflict and gendered parenting practices following the 2021 military coup. We included 731 adolescent girls aged 10–17 years enrolled in a girls' empowerment programme, recruited from Mandalay, Yangon and Tanintharyi Regions, Shan State and Kachin State. We co-developed a cross-sectional survey tool with a team of peer-researchers, conducting it between September 2023 and January 2024. Measures included physical and psychological violence, exposure to conflict-related stressors and gendered parenting practices. Logistic regression analyses tested associations between key variables. 80.6% of participants reported psychological violence and 49.8% reported physical violence in the past year. Participants reporting 3+ conflict-related stressors were more likely to report physical violence (aOR=2.19, 95%CI = 1.24-3.89, p = 0.007), participants reporting 1–2 stressors were more likely to report psychological violence (aOR=2.04, 95%CI = 1.09-3.77, p = 0.025). Higher gendered parenting scores were associated with physical (aOR=1.06, 95% CI = 1.03-1.09, p < 0.001) and psychological violence (aOR=1.07, 95%CI = 1.03-1.10, p < 0.001). Conflict may exacerbate gendered parenting and is associated with more parent/caregiver-perpetrated violence. These findings highlight the need for interventions addressing adolescent girls' unique health and wellbeing needs in conflict-settings.

**Data availability statement:** Due to the sensitive nature of this research, which was conducted in Myanmar following the military coup, the data cannot be made publicly available. The dataset contains information related to conflict exposure, experiences of violence, and mental health, and cannot be sufficiently anonymised to ensure participant confidentiality. Participants were assured during the informed consent process that their data would not be publicly shared. Sharing the data could also compromise the safety of local research partners. For these reasons, and in line with the ethical approvals granted by LSHTM and a local ethics board in Myanmar, the dataset will not be made available. Researchers may contact Isabelle Pearson (Isabelle.Pearson@lshtm.ac.uk) to discuss potential access on a case-by-case basis, pending ethical review. Alternatively, a non-author contact for data access requests is Ms Khin Yadanar Tun (khinyadanartun@colorfulgirls.org).

**Funding:** This work was supported by the LSHTM Doctoral Project Travelling Scholarship (to IP), the Bloomsbury Colleges (PhD studentship to IP), and the Millby Foundation (Grant: DONAT 15010 to CZ). The funders had no role in study design, data collection and analysis, decision to publish, or preparation of the manuscript.

**Competing interests:** The authors have declared that no competing interests exist.

## Introduction

Armed conflict can have long-term, life-altering effects on an individual's health and wellbeing, resulting not only in immediate harm but also long-term physical and psychological trauma [1]. The direct effects of conflict are often exacerbated by displacement, food insecurity, financial instability and the breakdown of social support systems and essential services such as healthcare [2,3].

Myanmar has a long history of protracted armed conflict and political turmoil, which intensified following the 2021 coup d'état. Myanmar's conflict and long-term political instability has driven endemic poverty, food insecurity and severe deficits in health and social services [4–6]. By 2024, conflict-related violence and displacement in Myanmar had resulted in more than three million internally displaced persons, with humanitarian assistance withdrawal leaving an estimated 18.6 million people deprived of access to essential services [7,8]. Besides, climate change has heightened the frequency and intensity of extreme weather conditions, like cyclones and heatwaves, further undermining livelihoods and infrastructure [7,9]. This intersection of environmental, economic and political crises overburdened Myanmar's already fragile state systems, with severe effects on the security, safety, education and physical and mental health of the people of Myanmar, especially youth [10].

### The effects of conflict on adolescent girls and young women: intersecting forms of violence

Globally, approximately 89.2 million adolescent girls aged between 10 and 17 years live in conflict-affected environments, representing one fifth of all adolescent girls worldwide [11]. Accurate data on conflict-exposed adolescents in Myanmar have not been collected, but according to the United Nations Office for the Coordination of Humanitarian Affairs [7], nearly 40% of internally displaced people in Myanmar are children. Repeatedly, studies demonstrate that adolescent girls and young women are disproportionately at risk of the severe direct and indirect health effects that occur during and after conflict, but they are often overlooked in international humanitarian responses [12].

As well as direct harm from combat, girls in conflict-affected settings are exposed to increased risks of sexual and gender-based violence, including forced marriage [13,14]. An estimated 39% of women and girls in conflict-affected settings have been exposed to physical or lifetime intimate partner violence according to a recent meta-analysis, and in 2023, 3,688 acts of conflict-related sexual violence were confirmed by the UN, though this is widely considered a significant undercount [15,16]. Collectively, studies show that exposure to sexual violence causes physical, as well as psychological, trauma, and a heightened risk of the likelihood of sexually transmitted infection and unwanted pregnancies [17]. For instance, globally, girls in conflict-affected settings are 20% more likely to marry as children than girls in non-conflict areas [11]. High rates of post-traumatic stress disorder (PTSD), depression, and anxiety have also been documented among adolescent girls in conflict areas, further compounded by stigma over sexual violence [18].

Concurrently, conflict also impedes critical health services, limiting access to sexual and reproductive healthcare, education and psychosocial services [19]. Economic uncertainty and school closures further limit adolescents' education and future job prospects [20]. Without these protective systems, adolescent girls are at increased risk of human trafficking, communicable disease, and social isolation, which, together with the erosion of essential services, can reinforce cycles of poverty and accelerate poor wellbeing and health outcomes [13,21].

In view of significant research into violence against women and girls in conflict settings, the unique vulnerabilities among adolescents require further exploration. Often, studies in conflict settings—and in general—fall short of addressing the intersecting risks faced by adolescents, who may be exposed to both intimate partner violence (IPV) as well as violence against children (VAC) [22]. IPV refers to physical, sexual or psychological abuse by a current or former partner and is often researched with respect to adult relationships [23]. On the contrary, VAC covers all forms physical and psychological violence, neglect, maltreatment, and exploitation including sexual abuse by either family or non-family members [23]. Adolescents aged between 12–18 years, however, exist at the intersection of these two types of violence such that they can experience IPV in early relationships whilst still being at risk of VAC by parents, caregivers or other adult persons [22].

Specific studies on VAC by parents towards their adolescent children are rare, but broader evidence including younger children indicates an association between conflict exposure and heightened parent-perpetrated VAC. For instance, Malcom et al. [24] reported that parents in conflict-affected areas versus non-conflict areas of Iraq were significantly more likely to use moderate and severe methods of corporal punishment. Similarly, research findings suggest associations between conflict exposure and more parental aggression in Palestine [25], child sexual abuse in Lebanon [26] and child maltreatment in Sri Lanka and Uganda [27]. Similarly, various studies in the United States have indicated that child maltreatment is elevated within families with one or both parents being members of the US military [14,28–30]. For example, Hisle-Gorman et al. [31] demonstrated that parental deployment was associated with child maltreatment against children aged three to eight, and for children of injured veterans in particular, there was a 24% rise in child maltreatment with each additional parental injury. Likewise, a meta-analysis carried out by Eltanamly et al. [32] revealed that war-exposed parents treated their children harshly and were less warm. That said, there is not a straightforward relationship between conflict exposure and parenting. Not all families with parents exposed to armed conflict respond with harshness at home. Various individual, familial and societal protective factors can reduce violence, which highlights the need for additional context-sensitive and nuanced research in this area.

Several theories help explain why exposure to armed conflict can escalate violence at home. Murphy et al. [33] synthesised published research and theoretical perspectives to create a conceptual framework for the drivers of violence against women and girls in conflict and post conflict settings. Their findings document risk factors for violence in six spheres: global, societal, institutional, community, interpersonal, and individual. Similarly, qualitative research from the Democratic Republic of Congo and Myanmar used an ecological framework approach to map out a network of intersecting risk factors and social norms that contribute to domestic violence in humanitarian settings [34]. Both studies highlight gender inequality as a key macro-level driver, which permeates into community and household dynamics, where conflict exacerbates hyper-masculine and patriarchal norms, weakens the rule of law, and normalises violence. Meanwhile, household stress—exacerbated by financial instability, food insecurity, and personal insecurity—increases the risk of domestic violence, particularly when individuals turn to negative coping mechanisms such as alcohol and substance abuse [33–35]. Although these theories focus on women and girls more broadly, they are relevant for adolescent experiences of violence, because of the shared risk factors between IPV and violence against children, and adolescents' risk of each form of violence [36].

There is a lack of research on parent or caregiver violence towards adolescents in Myanmar. A study by Aye et al. [37] surveyed 2,377 adults and found that 21.1% reported experiencing abuse during childhood, more commonly reported by women (29.8%) than men (12.4%). According to the 2016 Global School-Based Student Health Survey, 32.7% of students aged 13–17 years in Myanmar said they had been physically attacked in the past year, more common among boys (39.8%) than girls (26.3%) [38]. In another study, Nyan Linn et al. [39] analysed data from the 2015–2016 Myanmar

Demographic and Health Survey (DHS) and found that 44.5% of children aged 2–14 years had experienced corporal punishment, again with a higher prevalence among boys than girls. A study by Miedema & Kyaw [40] based on the same 2015–2016 DHS data identified that, among children ages 2–14 years, 18% reported abuse from a female caregiver, which was associated with women's past year exposure to physical or sexual IPV. Despite this, formal systems for preventing or responding to VAC in Myanmar are scarce and policy legislation around physical discipline of children remains ambiguous [41]. Although Myanmar adopted a new Child Rights Law in 2019 prohibiting all forms of physical punishment of children, such acts are still widely accepted in the home, day care, schools and alternative care settings [41,42]. Ultimately, efforts to prevent or address VAC in Myanmar remain limited, and research is needed on how the ongoing conflict may influence the prevalence of VAC, especially among adolescent girls, who face intersecting risks related to their age and gender.

### Gendered parenting practices and violence

"Gendered parenting", also known as differential parental socialisation, describes the different ways in which parents may treat their children based on their gender [43,44]. For instance, in focus group discussions with adolescent girls and young women in Myanmar, participants described how their parents were harsher and placed more restrictions on their freedoms than they did to their brothers [45]. The participants described how this unequal parenting had worsened since the 2021 coup, perhaps driven by factors such as heightened parental concern for their safety and growing financial stress at home. These gendered approaches to parenting are often rooted in beliefs about how boys and girls should behave, and they tend to reflect and reinforce inequitable gender norms [46]. For example, qualitative research from Uganda found that boys experienced greater freedom to play in their free time when compared to girls, who instead reported spending their free time learning gender-specific roles such as cooking [47]. Such practices not only affect childhood experiences but can also shape attitudes and behaviours that carry into adulthood.

Although gendered parenting has received limited attention in conflict-related research, it warrants a greater focus because of its potential role in helping us to better understand violence towards young women. Gendered parenting practices can lead to long-term disempowerment of daughters by reinforcing male dominance and traditional gender stereotypes [48]. Understanding this is especially important for conflict settings, where parents or caregivers (often hoping to protect their daughters' safety and reputations) may restrict their daughters' autonomy, mobility, and sometimes facilitate early or forced marriage [49]. The impacts of early and forced marriages, which include unwanted pregnancy, increased IPV and poorer mental health, are particularly harmful in humanitarian crises, where access to health care and support services is critically limited [50–52]. Evidence from conflict-affected settings, such as the Democratic Republic of Congo, suggests that caregiver's gender-equitable attitudes was associated with reduced risk of sexual abuse among adolescent girls, and reduced likelihoods of girls' acceptance of IPV [53]. In Myanmar, reports suggest that early and forced marriages have increased since the introduction of the 2024 forced conscription law, which exempts married women from military service [54].

### Gender inequality and societal norms

Gender inequality is prevalent across Myanmar, stemming from historically patriarchal traditions and societal norms that shape the expectations of girls and women from a young age. A key example is the concept of *hpon*, a widespread notion of male superiority over women under Theravada Buddhism in Myanmar [55]. Hpon perpetuates beliefs around women's impurity, positioning this impurity as a direct threat to men's *hpon*, which has contributed to women being excluded from leadership roles and restricted from certain religious spaces [55,56]. Similar gendered hierarchies exist among other religious and ethnic communities, and research has shown that women from minority backgrounds are disproportionately impacted by these inequalities [56]. Women and girls across the country risk stigma and consequences if they fail to uphold the gendered expectations of obedience and modesty that are expected of them, and adolescent girls are

especially impacted by these gendered norms due to their young age [55–57]. For example, from an early age, many girls are socialised to take on household responsibilities and gendered roles within the family to be considered a "good daughter", and—despite greater emphasis on girls' education in recent years—their schooling is still widely considered to be less important than boys', especially in financially insecure settings [58]. Furthermore, girls must navigate entrenched taboos around their sexuality and reproductive health and can face stigma following experiences of gender-based violence, restricting their ability to access information and support and putting their health and wellbeing at risk. These dynamics are further heightened during times of conflict, economic instability and displacement, which simultaneously exacerbate risk to girls' health whilst also weakening the systems that may otherwise protect them [58–60].

## Conceptual framework, aims and objectives

Fig 1 outlines the conceptual framework for this analysis, based on our previous qualitative research [45] and our review of the published literature. Fig 1 illustrates how the situation of armed conflict is reinforcing an existing system of gender inequality and societal norms. We believe that both the armed conflict and structural gender inequalities shape gendered parenting practices and parent/caregiver perpetrated violence. We also believe that gendered parenting practices may increase the prevalence of parent/caregiver perpetrated violence. Ultimately, we predict that both gendered parenting practices and parent/caregiver violence likely result in negative mental health and wellbeing outcomes for adolescent girls and young women. These health and wellbeing outcomes will be at greater risk due to the continued political instability, which is blocking the already-limited mental health and psychosocial support services, especially in the rural and conflict-affected areas [61].

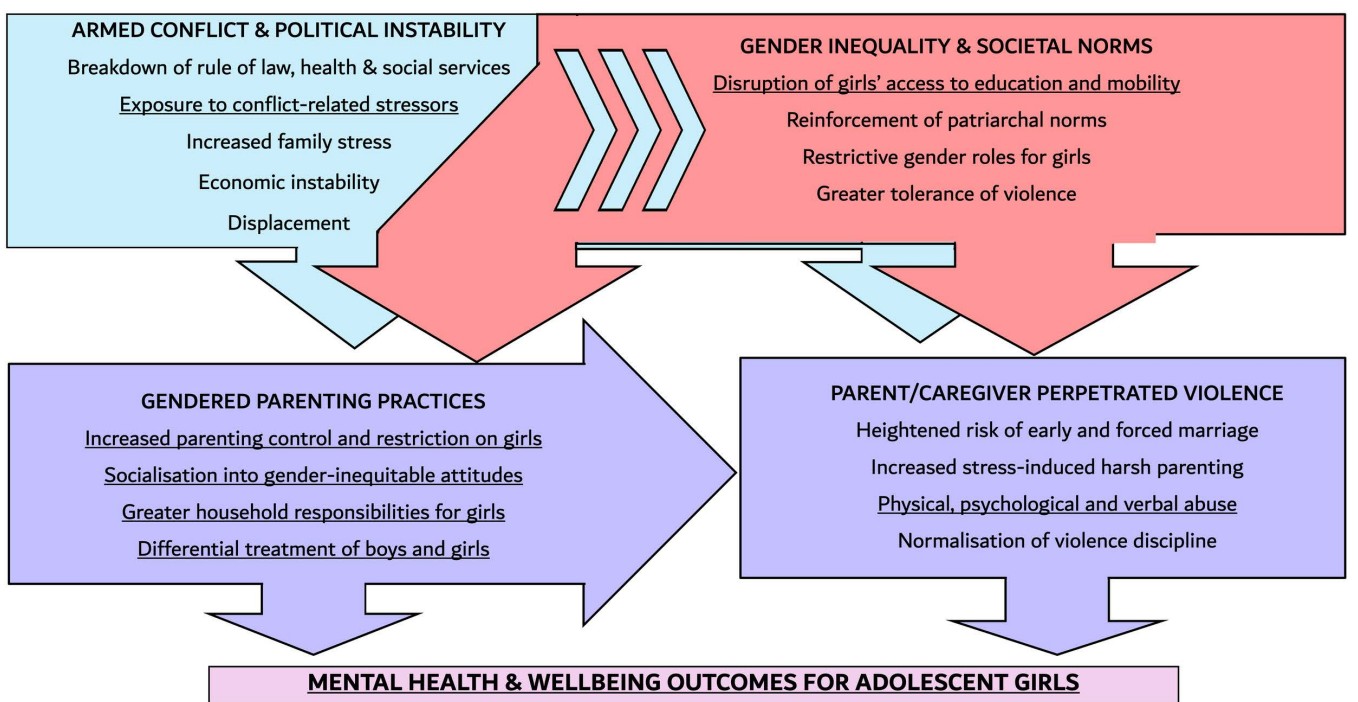

**Fig 1. Conceptual framework for our analysis, summarising hypothesised links between armed conflict, gender inequality, gendered parenting practices and parent/caregiver perpetrated violence among adolescent girls and young women.** The purple boxes represent domains shaped by both the armed conflict and gender inequality, to reflect their compounding influence.** highlights the data informing this study.

Despite the substantial risks of harm to young women and girls during conflict, to our knowledge, few studies have explicitly investigated the association between gendered parenting and violence against adolescent girls, and there have been none that consider this amidst the conflict in Myanmar. This study aims to measure the prevalence of violence by parents or caregivers among a sample of adolescent girls in Myanmar and examine associations between parents' use of violence and girls' experiences of gendered parenting following the 2021 military coup.

## Methods

### Ethics Statement

For participants under the age of 18 years, we collected written informed assent from all participants and either written informed parental consent or the written informed consent of a trusted adult on behalf of each participant. For participants aged 18 years or over, informed written consent was collected. Ethical approval for this research was received from the London School of Hygiene and Tropical Medicine Ethics Committee (ref 28458–2) and a local research ethics committee in Myanmar. Before enquiring about sensitive topics, participants were reminded of the option to skip any questions. All interviews were conducted in enclosed rooms where privacy could be maintained and all peer-researchers had prior experience researching sensitive topics, particularly violence. Breaks were offered throughout the course of the interview and participants were informed that they were able to withdraw their consent and end the interview at any time. Participation was voluntary and no monetary compensation was provided as the interviews were conducted as part of participants' ongoing GD programme schedule. Referral pathways were ensured through our partner organisation, GD.

### Inclusivity and global research

Additional information regarding the ethical, cultural, and scientific considerations specific to inclusivity in global research is included in the Supporting Information (S1 Checklist)

### Study design and participants

We conducted this study in collaboration with Myanmar-based NGO, Girl Determined (GD). GD supports girls and young women aged 11–25 years from low-income rural and urban settings across Myanmar. The two-year core programme, *Circles*, is an in-person weekly girls empowerment programme, facilitated by women hired and trained from local communities. *Circles* covers topics such as: self-confidence, assertiveness, ethical and religious discrimination, trafficking and safe migration practices, stress management, relationships (both romantic and friendships), drug awareness, goal-setting and sexual and reproductive health education. GD recruits hard-to-reach girls through sites such as camps for internally displaced people, churches, Buddhist nunneries and dormitories for girls from rural areas. Therefore, all girls enrolled in *Circles* generally represent girls from the lowest socio-economic quintiles in the regions of Myanmar that GD is active in. Due to GD's targeted outreach to nunneries and boarding houses, our sample included a high proportion of girls who were living away from home and enrolled in education, as these institutions provide education and accommodation to girls from low-income communities across Myanmar. At the time of survey, each participant was enrolled in GD's Circles programme and enrolment occurs at the site-level, for example, all girls aged 10–18 years who are residing at a selected boarding house will be invited to join. For more details about GD's work, please see Girl Determined [58].

The design and content of the survey was co-developed with six young, female peer-researchers from GD (EES, HJ, KL, NDO, NMS, ZMH), four of whom (EES, KL, NDO, ZMH) also co-developed and facilitated a prior qualitative study which helped inform the conceptual framework for this study see Pearson et al. [45] for additional details. The peer-researchers conducted the surveys in person, after receiving training on data collection methods and research ethics from a British doctoral student (IP) and a Myanmar research assistant and translator CVK. Five sites were chosen for data collection, which were the home states of the peer-researchers and thus safe and accessible to them during the data collection period. The sites were: Yangon Region, Mandalay Region, Kachin State, Shan State and Tanintharyi Region—all

sites where GD implements their empowerment, digital literacy and sports programmes. Within each region, we randomly selected sites where GD were already implementing the *Circles* programme. We randomly selected sufficient sites for each peer-researcher to conduct 125 surveys, whilst purposely excluding any that were unsafe due to conflict or weather. Two peer-researchers were based in Mandalay Region and one each in Yangon Region, Shan state, and Kachin Region and one peer-researcher conducted interviews across both Yangon and Tanintharyi Regions. The participant recruitment period lasted from 22/09/2023 until 16/01/2024. At each site, we invited every girl registered with GD (most of whom are aged 10–18 years) to complete a survey. The main reason girls declined to take part in the study was school exams.

## Procedures

The survey tool was co-developed with the peer-research team in September 2023 in Yangon which involved multiple participatory workshops where peer-researchers reviewed, refined and contributed to the adaptation of survey questions to ensure cultural and contextual appropriateness. Most survey questions are based on existing tools, some were also validated for use with adolescents in low-income settings. None of the questions we used were validated in our setting due to the scarcity of existing Myanmar-validated tools and our own time and resource constraints. During our co-production workshops, our survey tool was reviewed and developed by the research team to ensure it was culturally and contextually appropriate. CVK translated the survey tool, which was checked by bilingual members of GD staff. We conducted eight pilot interviews, and further developed the tool based on the pilot-results. For questions with Likert scale responses, a card was presented to participants with a visual representation of the scale using smiling faces and a green to red colour code.

The survey was conducted between September 2023 and January 2024. The peer-researchers read all questions out loud to the participants and then recorded their verbal responses into Kobo Toolbox using mobile phones. Each survey lasted approximately one hour and collected data on participant demographics, their living situation, social support, exposure to conflict, experiences of parent or caregiver perpetrated physical and psychological violence, mental health and hope. An in-depth analysis of the results pertaining to mental health and conflict exposure have been published elsewhere [62]

## Measures

**Conflict exposure.** To assess participants' conflict exposure, we combined four questions from Macksoud's [63] Child War and Trauma Questionnaire (Child subset) with three new questions tailored specifically to our participants' context. The new questions were co-developed from discussions with local experts. Participants were introduced to the topic with an explanation about the ongoing political conflict and then asked: "Thinking about between now and 2021, please answer yes or no to the following questions: 1) have you had to flee your home or move house because of the conflict, even if only for one night?; 2) have you had to move schools due to the conflict?; 3) have you been prevented from having access to your usual education due to the conflict?; 4) have you been separated from your parents/caregivers due to conflict?; 5) have you lost any family members due to the conflict?; 6) have you been directly exposed to armed combat (such as shelling, shooting, bomb explosion)?; 7) have you suffered any injury as a result of the conflict?". A conflict exposure score ranging from 0-7 was calculated based on affirmative responses to each question and respondents were then grouped based on their total number of conflict exposures, as a proxy measure for the severity of conflict exposure: no exposures, 1–2 exposures, and 3–7 exposures. A detailed analysis of conflict exposure results has been detailed elsewhere [62].

**Gendered parenting.** To measure gendered parenting, we used a proxy, subjective measure of the participants' perceptions of their parents' gendered parenting. To measure this, we developed a set of eight questions. The questions were co-developed with the peer-research team based on discussions around gendered parenting during our prior qualitative research and drawing from the questions around gendered social contexts from the Global Early Adolescent Survey [64,65]. As we could not compare participants' answers with their brothers, we could not measure gendered parenting directly. We asked participants with at least one brother how much they agreed or disagreed with the following

statements: 1) My parents allow my brother more freedom to spend time with his friends than me; 2) My parents trust my brother with more responsibility than me; 3) My brother receives more praise from our parents than me when we complete similar tasks/achievements; 4) My brother receives more money from our parents than me; 5) My parents think that household chores are more my responsibility than my brothers; 6) My parents expect me to marry at a younger age than my brother; 7) My parents expect me to finish school at a younger age than my brother; 8) My parents love my brother more than me. For each question, participants could answer that they strongly agreed, slightly agreed, slightly disagreed and strongly disagreed.

**Parent/Caregiver-perpetrated violence.** To measure experiences of physical and/or psychological violence from parents/caregivers we used a modified version of the International Society for the Prevention of Child Abuse and Neglect's (IPSCAN) Child-abuse Screening Tool for Children scale (ICAST-C) [66]. During our co-development workshops, we modified ICAST-C by reducing the number of questions and adding one question (Q12). We asked participants: "In the past year, how often have you experienced the following from a parent or caregiver: 1) shouted, yelled, or screamed at you very loudly; 2) insulted you, cursed at you or called you hurtful names; 3) threatened to hurt or kill you; 4) punched you, kicked you or hit you with a closed fist; 5) hit or slapped you with their hand; 6) hit you with an object; 7) shook you; 8) pushed you; 9) threw something at you; 10) cut you with a sharp object; 11) burnt you on purpose; 12) forced you to eat something unpleasant". Participants answered using a 7-point Likert scale ranging from "once a week or more" to "never in my life". For our analyses, we grouped participants into three groups for physical and psychological violence both together and separately: "frequent" if they reported any violent act once a month or more, "moderate" if they reported any violent act occurring less often than once a month but at least once a year, and "not in the past year" if they reported no violent acts in the past year or ever.

**Covariates.** To account for potential confounding factors, we also collected data on the following covariates: age (measured in years); school grade (based on the highest grade achieved); and whether the participants were living with their birth parents or not (binary variable: yes/no).

## Statistical analyses

We summarised key demographic and behavioural variables using descriptive statistics, stratified by geographical region and age group. We measured exposure to physical and psychological violence in the past year using binary indicators derived from participant responses to the violence-related questions. An affirmative response to any question categorised participants as exposed. For the gendered parenting scale, we coded responses numerically (−2 to +2, where disagreement indicated lower gendered parenting bias) and summed to generate a continuous gendered parenting score. Scores were summarised using means, medians, standard deviations, and ranges. We conducted binomial tests to compare agreement versus disagreement for each question. We assessed conflict exposure using a categorical variable representing three levels of exposure (none, 1–2 events, 3+events), with dummy variables created for logistic regression analyses. We assessed associations between violence and age group, as well as living arrangements, using $X^2$ tests and proportions for each subgroup. To analyse the association of gendered parenting and conflict exposure on violent parenting, we ran two logistic regression models: one for physical violence and one for psychological violence. Predictors included conflict exposure (dummy-coded), gendered parenting score, age, and whether participants lived with their birth parents. We calculated adjusted odds ratios (aORs) and 95% confidence intervals (CIs) to quantify associations. We conducted analyses in R Studio (version 4.3.0).

## Results

### Demographics

Table 1 presents the demographics of our study sample. Study participants included 731 adolescent girls and young women, aged 10–17 years. The overall median age was 14 years. Participants were living in villages and peri-urban areas in Mandalay (34.1%) and Yangon Regions (26.4%), as well as Kachin State (16.6%), Shan State (17.1%) and Tanintharyi

**Table 1. Participant demographics.**

| | Early Adolescents: 10–14 years (n = 467) | Mid-Late Adolescents: 15-17 years (n = 264) | Total (n = 731) |
|---|---|---|---|
| **Education** | | | |
| Currently enrolled in education | 464 (99.4%) | 253 (95.8%) | 717 (98.1%) |
| *Highest school grade:* | | | |
| Primary (G1–G4) | 121 (25.9%) | 2 (0.8%) | 123 (16.4%) |
| Secondary (G5-G11) | 343 (73.4%) | 259 (98.1%) | 602 (82.4%) |
| Other/Refused | 3 (0.6%) | 3 (1.1%) | 6 (0.8%) |
| **Living Situation** | | | |
| Live with parents | 270 (57.8%) | 85 (32.2%) | 355 (48.6%) |
| Live away from parents | 197 (42.2%) | 179 (67.8%) | 376 (51.4%) |
| Nunnery | 146 (31.3%) | 143 (54.2%) | 289 (39.5%) |
| With other family members | 23 (4.9%) | 10 (3.8%) | 33 (4.5%) |
| Boarding house/school | 11 (2.4%) | 15 (5.7%) | 26 (3.6%) |
| Other/Refused | 17 (3.6%) | 11 (4.2%) | 28 (3.8%) |
| **Area of Residence** | | | |
| Mandalay Region | 174 (37.3%) | 75 (28.4%) | 249 (34.1%) |
| Yangon Region | 124 (26.6%) | 69 (26.1%) | 193 (26.4%) |
| Shan State | 58 (12.4%) | 67 (25.4%) | 125 (17.1%) |
| Kachin State | 92 (19.7%) | 29 (11.0%) | 121 (16.6%) |
| Tanintharyi Region | 19 (4.1%) | 24 (9.1%) | 43 (5.9%) |
| **Ethnicity** | | | |
| Bamar | 209 (44.8%) | 67 (25.4%) | 276 (37.8%) |
| Kachin | 90 (19.3%) | 28 (10.6%) | 118 (16.1%) |
| Pa'O | 49 (10.5%) | 47 (17.8%) | 96 (13.1%) |
| Karen | 29 (6.2%) | 28 (10.6%) | 57 (7.8%) |
| Palaung | 28 (6.0%) | 28 (10.6%) | 56 (7.7%) |
| Other* | 62 (13.3%) | 66 (25.0%) | 128 (17.5%) |
| **Religion** | | | |
| Buddhist | 331 (70.9%) | 191 (72.3%) | 522 (71.4%) |
| Christian/Catholic | 132 (28.3%) | 72 (27.3%) | 204 (27.9%) |
| Muslim | 2 (0.4%) | 1 (0.4%) | 3 (0.4%) |
| Hindu | 2 (0.4%) | 0 (0.0%) | 2 (0.3%) |

*includes ethnic groups <25 participants: Chin, Danu, Dawei, Hindu, Intha, Kayah, Kayan, Khmer, Lahu, Lisu, Mon, Mro, Muslim, Rakhine, Shan, Taungyo and mixed ethnicities.*

Region (5.9%). Approximately half of participants (51.4%) were living away from their parents at the time of the survey, which was more common among mid-late adolescents (67.8%) than early adolescents (42.2%). Those living away from their parents were most often living at Buddhist nunneries (39.5%). Overall, 98.1% of our participants were enrolled in some form of education at the time of the survey.

**Violence perpetrated by a parent or caregiver**

A total of 715 participants chose to answer the questions on their experience of parent/caregiver-perpetrated psychological and physical violence, hereafter referred to as caregiver-perpetrated violence. The participants most often listed parents as their primary caregivers. Non-parent caregivers were primarily other family members but also included Nuns and

boarding house teachers for some of the girls who were living away from home. Fig 2 depicts an overview of the results and Table 2 provides a detailed breakdown of the different physical and psychological violence types and their reported frequency.

Of the 715 participants, 80.6% experienced at least one form of psychological violence in the past year and 49.8% experienced at least one form of physical violence. Both past-year physical and psychological violence were more frequently reported by the early adolescents, but this difference was only statistically significant for physical violence (53.1% of early-age adolescents vs. 44.7% of mid-to-late age adolescents; $X^2 = 4.3$, p = 0.038). For psychological violence, there was no significant difference between age groups (83.4% early adolescents vs. 80.6% of mid-late adolescents, $X^2 = 0.68$, p = 0.411). The most frequently reported type of psychological violence was being shouted, screamed or yelled at loudly and the most frequent physical violence was to be hit with an object (Fig 2).

Caregiver perpetrated psychological violence was reported more often by participants who lived either with their parents or another family member (51.5%) versus those living away from family (48.5%). However, this difference was not significant ($X^2 = 3.1$, p = 0.076). Conversely, caregiver-perpetrated physical violence in the past year was more often reported by participants who lived away from family members (51.1%) compared to those living with family members (48.9%), however, this difference was not statistically significant ($X^2 = 0.38$, p = 0.540). It should

### Gendered Parenting

A total of 514 participants who had brothers chose to respond to the questions about gendered parenting. Fig 3 provides a summary of the percentage of participants who agreed or disagreed with the eight statements, and more detailed results are presented in Table 3. The mean gendered parenting score was -0.95 (SD = 7.11). Since this score is close to zero, it suggests that participants generally gave neutral responses, with a slight tendency toward disagreeing with statements that their parents showed favouritism to their brothers.

Table 3 also includes the results of binomial tests conducted for each of the eight gendered parenting statements. These tests measured whether significantly more participants agreed or disagreed with each statement. The results show that participants were significantly more likely to *agree* with certain statements suggesting unequal treatment

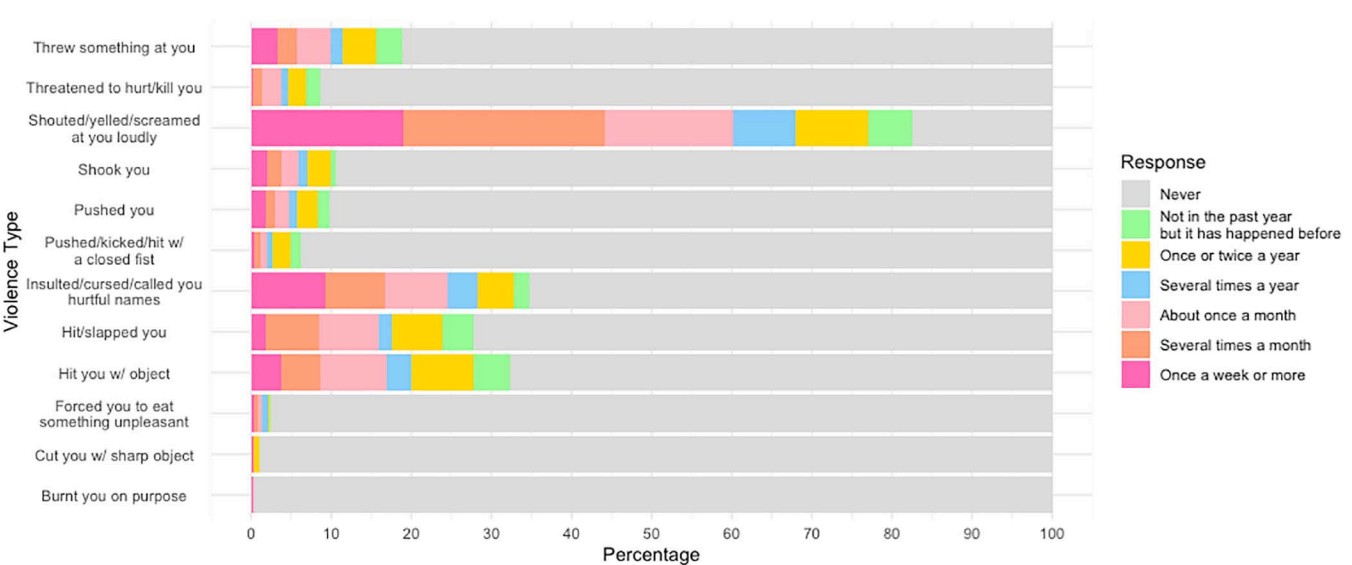

**Fig 2. Graph to show percentage of participants reporting different violence types (n = 715).**

**Table 2. Prevalence of physical and psychological violence types (n = 715) also be noted that some of the girls who lived away from family members still listed parents as their primary caregiver in the survey.**

| | Never | Before, but not in the past year | Once or twice per year | Several times per year | About once per month | Several times per month | Once per week or more |
|---|---|---|---|---|---|---|---|
| **Psychological Violence** | | | | | | | |
| Shouted, yelled, or screamed at you very loudly | 122 (17.5%) | 39 (5.6%) | 63 (9.0%) | 54 (7.7%) | 113 (16.2%) | 175 (25.0%) | 133 (19.0) |
| Insulted you, cursed at you or called you hurtful names | 456 (65.2%) | 14 (2.0%) | 32 (4.6%) | 25 (3.6%) | 55 (7.9%) | 52 (7.4%) | 65 (9.3%) |
| Threatened to hurt or kill you | 639 (91.4%) | 12 (1.7%) | 16 (2.3%) | 6 (0.9%) | 17 (2.4%) | 8 (1.1%) | 1 (0.1%) |
| **Physical Violence** | | | | | | | |
| Hit you with an object | 473 (67.7%) | 32 (4.6%) | 54 (7.7%) | 22 (3.1%) | 57 (8.2%) | 35 (5.0%) | 26 (3.7%) |
| Hit or slapped you with their hand | 505 (72.2%) | 27 (3.9%) | 44 (6.3%) | 12 (1.7%) | 52 (7.4%) | 46 (6.6%) | 13 (1.9%) |
| Threw something at you | 567 (81.1%) | 22 (3.3%) | 29 (4.1%) | 11 (1.6%) | 29 (4.1%) | 17 (2.4%) | 23 (3.3%) |
| Shook you | 625 (89.4%) | 5 (0.7%) | 20 (2.9%) | 8 (1.1%) | 14 (2.0%) | 13 (1.9%) | 14 (2.0%) |
| Pushed you | 631 (90.3%) | 10 (1.4%) | 18 (2.6%) | 7 (1.0%) | 12 (1.7%) | 8 (1.1%) | 13 (1.9%) |
| Punched/kicked/hit you with a closed fist | 655 (93.7%) | 10 (1.4%) | 16 (2.3%) | 4 (0.6%) | 6 (0.9%) | 5 (0.7%) | 3 (0.4%) |
| Forced you to eat something unpleasant | 682 (97.6%) | 1 (0.1%) | 1 (0.1%) | 6 (0.9%) | 3 (0.4%) | 3 (0.4%) | 3 (0.4%) |
| Cut you with a sharp object | 692 (99.0%) | 0 (0.0%) | 5 (0.7%) | 0 (0.0%) | 0 (0.0%) | 1 (0.1%) | 1 (0.1%) |
| Burnt you on purpose | 697 (99.7%) | 0 (0.0%) | 0 (0.0%) | 0 (0.0%) | 1 (0.1%) | 0 (0.0%) | 1 (0.1%) |

compared to their brothers. For example, 74.3% agreed that they had less freedom to see friends, 68.3% said they received less praise for similar tasks and 62.8% said they were assigned more chores, all of which were statistically significant (p < 0.001). On the other hand, participants were significantly more likely to *disagree* with statements suggesting that they had to marry earlier than boys (85.8% disagree, p < 0.001) or drop out of school earlier (78.4% disagree, p < 0.001). For statements about receiving lower allowances, being trusted less, or feeling less loved, there were no significant differences (p > 0.05).

Association between perceived gendered parenting, exposure to armed conflict and caregiver-perpetrated violence

The prevalence of conflict exposure and the types of conflict that the study population were exposed to have been detailed elsewhere [62]. In the present study, we conducted a multiple logistic regression analysis to test for associations between the number of conflict exposures and gendered parenting scores on caregiver-perpetrated violence perpetrated in the past year, whilst also accounting for participants' age and living status. The results are presented in Table 4.

The analysis presented in Table 4 found that participants who reported exposure to three or more conflict-related stressors were twice as likely to report physical violence in the past year compared to those without any conflict exposure (adjusted OR=2.19, 95% CI = 1.24-3.89, p = 0.007). The analysis also found that participants who scored higher on the gendered parenting scale were more likely to report physical violence, as each additional point on the scale was associated with a small but significant increase in the likelihood of experiencing violence (adjusted OR=1.06, 95% CI = 1.03-1.09, p < 0.001). Younger participants were also more likely to report physical violence, with the odds decreasing as age increased (adjusted OR=0.83, 95% CI = 0.73-0.95, p = 0.007). However, whether or not a participant lived with their birth parents was not significantly associated with reporting physical violence (p = 0.879).

Participants who reported exposure to 1–2 conflict-related stressors were twice as likely to report psychological violence in the past year when compared to those who did not report any exposure to conflict (adjusted OR=2.04, 95% CI = 1.09-3.77, p = 0.025). In addition, higher scores on the gendered parenting scale were also associated with an

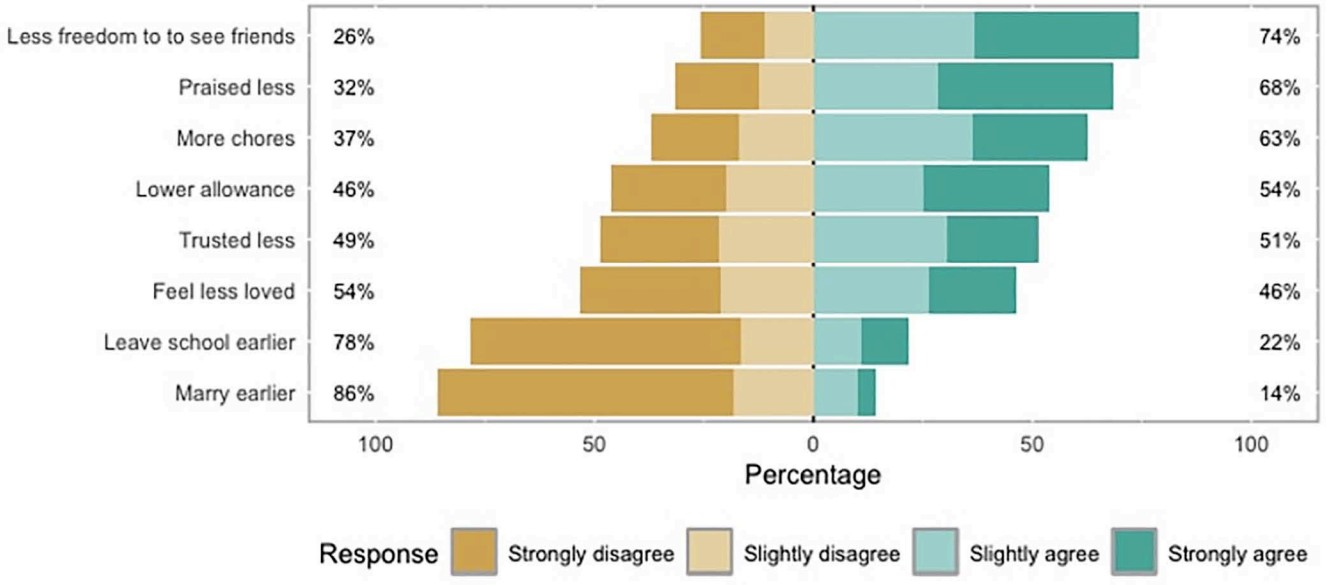

**Fig 3. Percentage of participants (n = 514) who agreed or disagreed with statements comparing their experiences of gendered parenting to how they perceive their brother(s) to be parented.** Agreement (right-hand side) and disagreement (left-hand side) are shown across four response categories: strongly disagree, slightly disagree, slightly agree, strongly agree.

**Table 3. Results of a binomial test to compare the proportion who agree with those who disagree for each of the gendered parenting questions (n = 514).**

| Question | Percentage Agree | Percentage Disagree | Significance |
|---|---|---|---|
| *My parents allow my brother more freedom to spend time with his friends than me* | 74.3 | 25.7 | p < 0.001 |
| *My brother receives more praise from our parents than me when we complete similar tasks/achievements* | 68.3 | 31.7 | p < 0.001 |
| *My parents think that household chores are more my responsibility than my brothers* | 62.8 | 37.2 | p < 0.001 |
| *My brother receives more money from our parents than me* | 53.7 | 46.3 | 0.103 |
| *My parents trust my brother with more responsibility than me* | 51.4 | 48.6 | 0.566 |
| *My parents love my brother more than me* | 46.5 | 53.5 | 0.123 |
| *My parents expect me to finish school at a younger age than my brother* | 21.6 | 78.4 | p < 0.001 |
| *My parents expect me to marry at a younger age than my brother* | 14.2 | 85.8 | p < 0.001 |

increased risk of psychological violence. While the increase per point was relatively small, it remained statistically significant (adjusted OR=1.07, 95% CI=1.03-1.10, p<0.001). By contrast, there was no significant relationship between psychological violence and either the participant's age (p=0.976) or whether they were living with their birth parents (p=0.274).

## Discussion

This study investigated the exposure of adolescent girls and young women across Myanmar to armed conflict, parent/caregiver perpetrated violence and gendered parenting practices. The findings presented here help contribute to a critical gap in the literature, not only on violence against women and girls in Myanmar, but also in conflict-affected settings more

**Table 4. Multiple logistic regression results for past year parent and caregiver perpetrated physical and psychological violence (n = 514).**

| Physical Violence | | | |
|---|---|---|---|
| Exposure | aOR | 95% CI | p-value |
| 1–2 conflict exposures | 1.05 | 0.62-1.79 | 0.848 |
| 3–7 conflict exposures | 2.19 | 1.24-3.89 | 0.007** |
| Gendered Parenting Score | 1.06 | 1.03-1.09 | <0.001*** |
| Age | 0.83 | 0.73-0.95 | 0.007** |
| Living with birth parents | 0.97 | 0.64-1.46 | 0.879 |
| Psychological Violence | | | |
| Exposure | aOR | 95% CI | p-value |
| 1–2 conflict exposures | 2.04 | 1.09-3.77 | 0.025* |
| 3–7 conflict exposures | 1.82 | 0.94-3.53 | 0.075 |
| Gendered Parenting Score | 1.07 | 1.03-1.10 | <0.001*** |
| Age | 1.00 | 0.84-1.18 | 0.976 |
| Living with birth parents | 1.33 | 0.80-2.24 | 0.274 |

aOR: adjusted odds ratio; * = p<0.05; ** = p<0.01; *** = p<0.001

Note: reference categories are no conflict exposures and not living with birth parents. Gendered parenting score and age are continuous variables

generally. Notably, this study considers how gendered parenting practices can be both exacerbated during conflict-related insecurity and a risk factor for parent/caregiver-perpetrated violence.

Parent/caregiver-perpetrated violence against children (VAC) is driven by structural, social and economic risk factors, many of which can be exacerbated in times of conflict. In this study, we used an ecological framework to highlight multiple levels of risk, such as inequitable gender norms and household stress [67,68]. While conflict exacerbates many of the pathways to parent/caregiver-perpetrated violence, including through displacement, trauma and financial stress, conflict is not a sole risk factor for VAC [17]. In Myanmar, structural factors such as widespread poverty and acceptance of gender inequitable norms, as well as cultural understandings of violence's role in effective parenting likely intersect with conflict-related stressors to heighten the risk of VAC. This study identified the alarming findings that over 80% of our study participants reported one or more past-year experiences of psychological violence, and almost 50% reported at least one form of physical violence over the same period. Notably, reports of violence were significantly more common among younger adolescents. By comparison, Linn et al.'s [39] cross-sectional analysis of Myanmar's DHS data (from 2015-2016) measured the prevalence of physical corporal punishment to be 44.5% among children aged 2–14 years. Though our study is not directly comparable to that of Linn et al., the finding that 49.8% of the adolescent girls experienced physical violence in the past year suggests increasing trends in caregiver-perpetrated violence subsequent to the DHS survey. According to Linn et al. [39], 39.6% of the girls reported moderate corporal punishment and 10.9% reported severe corporal punishment. However, Linn et al.'s study only included children aged 2–14 years, as opposed to our study of 10–17-year-olds, which may explain our higher prevalence. Furthermore, our study group consisted of individuals from communities that represent some of the lowest socioeconomic quintiles within Myanmar, which may explain the higher prevalence of violence due to poverty-related risk factors for violence against children [69]. These findings highlight the need for age-specific and sociodemographic-targeted interventions to address household violence that can be implemented in conflict-affected settings, particularly in marginalised communities.

The heightened insecurity and armed conflict that followed Myanmar's 2021 military coup may help to explain the increase in violence reported since the 2015–16 DHS. While conflict had been ongoing prior to the coup, our findings did indicate that exposure to more conflict-related stressors was associated with increased reports of parent/caregiver violence. These findings align with those from other conflict-affected contexts, such as Sri Lanka and Northern Ireland, which

have shown that exposure to political violence and displacement increases family stress and caregiver trauma, potentially leading to greater use of parent/caregiver violence [27,70]. In particular, our data indicate that the adolescent girls more severely affected by conflict (measured by higher numbers of conflict-related stressors, including exposure to active conflict, displacement, family deaths, personal injuries and disruption to schooling) also reported significantly higher levels of parental physical violence. Overall, these findings align with global evidence that suggests high levels of conflict exposure (such as experiencing direct violence, displacement, or losing family members) can create severe household stress and increase the chances of parent/caregiver violence [27,71–74].

However, this relationship differed with regards to the number of conflict-related stressors and the type of parent or caregiver violence: as moderate reports of conflict-related stressors (1–2 stressors) were significantly associated with higher reports of psychological violence, whereas higher levels of conflict-related stressors (3–7 stressors) were not. This finding suggests that the relationship between conflict exposure and household violence is not a simple linear relationship, and that other factors may mediate this interaction. This finding might also reflect how different types of conflict-related stressors were weighted equally in our analyses, meaning that our proxy measure for the severity of conflict exposure may not accurately capture the severity based on the types of stressors reported, e.g., in our analysis, losing a family member was weighted the same as conflict-disrupted education. Future research and analysis should aim to develop a more accurate weighted tool to explore the links between different types of conflict-related stressors and household violence in this context. Likewise, research should seek to collect temporal data, including on conflict-related stressors such as household stress, displacement, food insecurity and financial instability, to better understand the relationship between conflict and household violence.

Interestingly, in our sample, living with birth parents versus elsewhere was not associated with differing levels of parent or caregiver violence and as noted, many participants who lived away from home still named parents as their primary caregiver. Future research should seek to identify who specifically perpetrated the violence, rather than categorising them broadly as parents or caregivers. Despite this ambiguity, given that many of the respondents had been living away from home for the past year, it is very plausible that non-parent caregivers are also perpetrating violence towards participants, which warrants further investigation.

When analysing parenting practices, we identified a significant positive association between higher gendered parenting scores and parent/caregiver perpetrated physical and psychological violence, suggesting that gender norms exacerbate the risk of violence for girls. These findings corroborate the qualitative findings from our previous research with adolescent girls in Myanmar, who described feeling like their brothers had priority and were treated beneficially within their families [45]. The present study's findings on perceived gendered parenting among our study population also support the findings of Kågsten et al.'s [48] systematic review, which identified global evidence of parental control and restrictions on girls, including limits on their activities and mobility. However, unlike Kågsten et al., we did not find strong evidence that gendered parenting resulted in regulation on girls' education, as 78.4% of our population disagreed with the notion that they would have to drop out of school earlier than their brothers. Our findings that both higher levels of conflict exposure and gendered parenting scores were significantly related with increased caregiver violence, indicate that there may be a causal pathway linking the three variables; specifically, conflict could act as a moderator, exacerbating the association between gendered parenting practices and caregiver violence.

In our study, the dimension of gendered parenting with the strongest agreement level (74.3%) was the notion that the participants' brothers were granted comparatively more freedom to spend time with their friends. One theory that could help explain this finding is Endendijk's [75] Gendered Family Process Model, which posits that one reason for gendered parenting is that parents attribute boys' risky behaviour to inborn characteristics, whereas similar behaviour in girls is viewed as situationally driven. Endendijk's model suggests that parents may be more likely to restrict their daughter's risky behaviour by reducing their interaction with environments that they consider risk-inducing [75]. In light of our findings, this theory could help explain why girls felt their freedoms were restricted when compared to their brothers, especially as

these girls are living in high-risk environments due to the ongoing armed conflict, which may encourage more restrictive parenting practices.

Our findings indicate that 62.8% of participants agreed with the statement that household chores were considered more their responsibility than their brother's. When considered alongside the 74.3% who agreed they had less freedom for personal enjoyment and social interaction compared to their brothers, our results suggest that girls in this setting had much less autonomy. Likewise, girls reported receiving less praise in comparison to their brothers, which may contribute to a sense of being undervalued by their families, despite their higher contribution to chores and time spent at home.

In addition to the negative effects that these forms of gendered parenting may have on young women, such practices may also have implications for boys and young men. Although beyond the scope of this study, it is important to acknowledge the impact of socialising boys and young men to gender-inequitable views and behaviours early in life. For instance, Halpern et al. (2016) found that parents' behaviours strongly predicted children's attitudes, with mothers' actions in particular having significant influence on both their sons' and daughters' understanding of gender roles. Similarly, Bumpus et al. [76] found that parents with traditional gender role attitudes were less likely to grant their daughters decision-making input than their sons, potentially reinforcing male superiority and perpetuating gender inequalities. While these findings suggest the importance of interventions promoting gender-equitable parenting, such efforts are especially challenging in conflict-affected settings, where parents must prioritise their family's immediate physical safety, potential displacement and day-to-day survival. Therefore, in such contexts, more feasible intervention pathways should be prioritised. In our study setting, many girls remain accessible through schools or programmes such as Girl Determined, and therefore interventions aiming to equip girls and young women with the tools and confidence to advocate for themselves within their families may be more feasible than targeting parents directly. This approach would facilitate generational change, equipping the next generation of mothers with more gender equitable perspectives, whilst also ensuring that they do not continue to be overlooked in resource-poor settings. Likewise, in contexts where schools remain open (such as in our study sites, where 98% of participants were still enrolled in some form of education) school-based interventions might offer a viable opportunity to reshape boys' perceptions of gender roles. By emphasising equality and respect towards women, such interventions could help to counteract the superiority messages that boys have internalised throughout their upbringing.

## Benefits and limitations

This study offers a novel insight into the lives of adolescent girls in conflict-affected Myanmar and provides a unique analysis of the intersections of conflict, violence and gendered parenting practices, which to the best of our knowledge has not been investigated before in this setting. Furthermore, this data provides important evidence for just some of the many negative impacts that Myanmar's protracted conflict is having on adolescent girls, and in doing so advocates for an often-overlooked demographic. That said, besides the specific limitations related to the use of proxy measures, there are a few additional considerations when interpreting the results of this study. First, as mentioned, none of the survey tools were officially validated with adolescent girls in this setting. However, our co-production approach and in-depth co-production workshops—where we dedicated time to developing the survey tools in collaboration with young peer-researchers who were previously among the study population demographic themselves—significantly increased our confidence in the tools. We believe that although not validated in our setting, our co-production approach allowed for comprehensive tool-development given our time and resource constraints.

For the violence-related questions especially, we relied on self-reported data, which are at risk of recall bias. In particular, telescoping bias, where participants may recall events as occurring more recently or further in the past than they actually occurred [77]. Furthermore, studies have shown that participants with post-traumatic stress disorder show bias towards recalling negative information which could be relevant to our study population [78]. To mitigate this recall bias, we limited our violence analysis to only consider violent acts that happened in the previous year. However, we did not assess the severity of the violence experienced, meaning that all forms of violence were weighted equally in our analyses,

regardless of the type of act. This approach may have overlooked important distinctions between different types of violence and their relative severity. Additionally, as this survey was conducted in 2023, we allowed for a two-year window of conflict exposures since the 2021 military coup, meaning that the conflict data is at slightly higher risk of recall bias [79]. Finally, our measure of parent/caregiver-perpetrated violence did not also enquire about demonstrations of love, support or care, meaning our measures of violence are being considered in isolation of any broader changes in parent/child relationships.

Our gendered parenting scale is limited in that we did not enquire with the parents or the brothers of our participants, meaning that our conclusions are based wholly on the subjective perceptions of the participants, introducing a range of self-reporting biases such as perception bias, recall bias, confirmation bias and social desirability bias. Due to the risk of multiple forms of bias in our gendered parenting measure, the limited generalisability of our findings should be noted, and we acknowledge that our findings may not fully capture the broader family dynamics or the interactions of parents and brothers. Further research into gendered parenting practices should include multiple perspectives and additional methods such as observations to help triangulate the results. Moreover, while we aimed to use random sampling, the ongoing insecurity and instability across Myanmar sometimes requires last-minute revision of survey sites, meaning random sampling was not always possible. Participants were invited who were current beneficiaries of GD's empowerment programme, which limits the generalisability of findings to girls outside this programme. It should also be noted that the participants of this study had already received training and education on topics such as gender equality and violence, which may have influenced their responses to questions around these topics. Furthermore, the region-based sampling could have created clustering effects by region. Since we only included five regions with modest sample sizes in each, we did not use multi-level modelling or adjustments for clustering affects. Finally, as with all cross-sectional studies, we cannot investigate the causal pathways between our variables or make robust predictions about mediation or moderation. That said, we can conclude that there are statistical associations between conflict, violence and gendered parenting practices within our study population, and therefore future research into tailored violence-prevention interventions should take this into account.

Furthermore, while this study offers important insights into the quantitative associations between key variables, the complexity of adolescent girls' experiences highlights the need for complementary qualitative research to better understand the nuances behind these intersecting risk factors. While outside the scope of this study, qualitative data from parents and caregivers would offer a further understanding of gendered parenting practices and the use of violence within the home, including why these behaviours may be exacerbated during periods of instability and uncertainty. Such research could also help to identify contextually appropriate targets for future interventions to protect adolescent girls from these risk factors.

## Conclusion

Our findings highlight a significant overlap between armed conflict, gendered parenting practices, and parent or caregiver perpetrated violence. Given the documented risks of abuse and conflict on the health and wellbeing of adolescent girls, groups working in conflict-affected settings should be alerted to both exposures when considering health and other interventions. Specifically, the prevalence of parent/caregiver perpetrated violence and its strong association with conflict and parental gender norms demands responses in these settings that consider risks at multiple levels of the socioecological framework. The finding that a high prevalence of physical violence was found among younger adolescents in particular highlights the necessity for age-specific responses. Lastly, while armed conflict poses immediate and serious risks, our results highlight the potentially long-term consequences for girls and young women, impacting their long-term safety, autonomy and overall health and wellbeing. Interventions should ensure the physical safety and psychological wellbeing of girls and young women, whilst also engaging boys and young men to ensure they are not reflecting their parents' or society's negative gendered attitudes. These findings highlight the need for adaptable, multi-sectoral interventions that reflect the complexity of conflict-affected settings and girls' wellbeing, which requires activities related to violence prevention, mental health support, empowerment, and the shifting of harmful gendered norms within families and communities.

Programmes such as Circles offer a valuable opportunity to support girls and young women, not only directly, but also through targeted outreach to parents and caregivers. This study's findings may serve as a useful advocacy tool to encourage caregivers to reflect on their parenting practices and to inform intervention strategies that promote safer and more gender-equitable home environments.

## Supporting information

**S1 Checklist. Inclusivity in global research.**
(DOCX)

## Acknowledgments

The authors would like to thank all of the staff at Girl Determined for their commitment, support and guidance throughout this study, especially Khin Yadanar Tun, Moon Pan and Arkar. We would also like to thank Dr San Shwe, Dr François Tainturier, Aileen Thomson and Dr Giulia Greco for their guidance.

## Author contributions

**Conceptualization:** Isabelle Pearson, Elaine Chase, Cing Van Kim, Nang Ma San, Hkawn Ja, Ei EI Soe, Khin Lae, Nandar Oo, Zin Mar Hlaing, Brooke Zobrist, Cathy Zimmerman, Meghna Ranganathan.

**Data curation:** Isabelle Pearson, Cing Van Kim, Nang Ma San, Hkawn Ja, Ei EI Soe, Khin Lae, Nandar Oo, Zin Mar Hlaing.

**Formal analysis:** Isabelle Pearson, Elaine Chase, Cing Van Kim, Nang Ma San, Hkawn Ja, Ei EI Soe, Khin Lae, Nandar Oo, Zin Mar Hlaing, Meghna Ranganathan.

**Funding acquisition:** Isabelle Pearson, Brooke Zobrist, Cathy Zimmerman, Meghna Ranganathan.

**Investigation:** Isabelle Pearson, Elaine Chase, Cing Van Kim, Nang Ma San, Hkawn Ja, Ei EI Soe, Khin Lae, Nandar Oo, Zin Mar Hlaing, Brooke Zobrist, Cathy Zimmerman, Meghna Ranganathan.

**Methodology:** Isabelle Pearson, Elaine Chase, Brooke Zobrist, Cathy Zimmerman, Meghna Ranganathan.

**Project administration:** Isabelle Pearson, Cing Van Kim, Nang Ma San, Hkawn Ja, Ei EI Soe, Khin Lae, Nandar Oo, Zin Mar Hlaing, Brooke Zobrist, Cathy Zimmerman, Meghna Ranganathan.

**Resources:** Isabelle Pearson, Brooke Zobrist, Cathy Zimmerman, Meghna Ranganathan.

**Software:** Isabelle Pearson.

**Supervision:** Elaine Chase, Brooke Zobrist, Cathy Zimmerman, Meghna Ranganathan.

**Validation:** Isabelle Pearson, Elaine Chase, Cing Van Kim, Nang Ma San, Hkawn Ja, Ei EI Soe, Khin Lae, Nandar Oo, Zin Mar Hlaing, Brooke Zobrist, Cathy Zimmerman, Meghna Ranganathan.

**Visualization:** Isabelle Pearson.

**Writing – original draft:** Isabelle Pearson.

**Writing – review & editing:** Elaine Chase, Cing Van Kim, Nang Ma San, Hkawn Ja, Ei EI Soe, Khin Lae, Nandar Oo, Zin Mar Hlaing, Brooke Zobrist, Cathy Zimmerman, Meghna Ranganathan.

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
