## [Decision Letter · Decision Letter 0]

18 Jun 2025

PGPH-D-25-00963

Conflict, caregiver violence and gendered parenting: A cross-sectional study among adolescent girls and young women in Myanmar

Dear Dr. Pearson,

Thank you for submitting your manuscript to PLOS Global Public Health. After careful consideration, we feel that it has merit but does not fully meet PLOS Global Public Health’s publication criteria as it currently stands. Therefore, we invite you to submit a revised version of the manuscript that addresses the points raised during the review process.

We look forward to receiving your revised manuscript.

Kind regards,

Zahra Zeinali, MD MPH DrGH (c)

Academic Editor

Journal Requirements:

Additional Editor Comments (if provided):

Reviewers' comments:

Reviewer's Responses to Questions

**Comments to the Author**

1. Does this manuscript meet PLOS Global Public Health’s publication criteria?

Reviewer #1: Yes

Reviewer #2: Yes

2. Has the statistical analysis been performed appropriately and rigorously?

Reviewer #1: Yes

Reviewer #2: Yes

3. Have the authors made all data underlying the findings in their manuscript fully available (please refer to the Data Availability Statement at the start of the manuscript PDF file)?

Reviewer #1: No

Reviewer #2: No

4. Is the manuscript presented in an intelligible fashion and written in standard English?

Reviewer #1: Yes

Reviewer #2: Yes

Reviewer #1: Abstract

• The methods section should specify the Country, not only the regions and state

• The conclusion could be more specific by explicitly stating the study’s implications.

Introduction

1. Grammar and sentence structure should be revised throughout the paper for clarity and readability.

2. The clarity of sentences should be revisited to ensure grammatical accuracy.

Results

Why there is only 715 from total of 731 respondents answered the questions on their experience of

parent/caregiver-perpetrated psychological and physical violence, while actually there is an answer option of: Never, meaning not all of them respond to this question are the victim of violence?

References

Add some references to enhance the literature review:

Caregiver parenting and gender attitudes: Associations with violence against adolescent girls in South Kivu, Democratic Republic of Congo

Gendered and transactional relationships: Children's experiences of being parented and perspectives on their relationships with their parents in Uganda

Reviewer #2: This is a very timely and needed study carefully conducted in an active conflict situation. The authors and their counterparts should be applauded for conducting this study in such a fragile situation. It is a well-written paper in excellent academic english language.

Title: I am a bit concerned about this components of the title: A cross-sectional study among adolescent girls and young women in Myanmar

As of now, it is implied that the study is representative of Myanmar. However, from methodology, it is clear that the sample frame is adolescent girls and young women who are participating in the circle programme, not the total adolescent and girl population of Myanmar.

Suggest indicating this in the title. Ex: A cross-sectional study among adolescent girls and young women participating in a weekly girls empowerment programme in Myanmar

This is especially true, as the study attempts to provide a population proportion with a confidence interval, however. The sampling method, as I understand, is not a probabilistic sampling method. While I totally appreciate that would be not feasible, authors need to be careful that they are not over-generalizing their sample to the total of the adolescents and girls in Myanmar.

It would be helpful to indicate how similar and different are girls and women in circle sites? How long has these circles been running? Has there been any awareness raising or empowerment that happened in these circles before the study? In such cases, the sampled population is likely to be different from the average girls and women population, right?

Also please see my comment about a map demonstrating sampling sites so that the reader gets a clear idea about the generalizability of the sample.

It would be good to mention about the current systems or lack thereof for the prevention or management of parent/caregiver violence in Myanmar, including support networks. Are there any systems that cater to the parents and caregivers? Unlikely, but I think it is essential to highlight this. I was wondering if you should indicate health/mental health and psychosocial support systems (or lack thereof) in your conceptual framework.

Out of the three major components of the conceptual framework, all except Gender Inequality and Societal Norms have nicely been addressed in the sections prior to that. Suggest adding a subsection titled Gender Inequality and Societal Norms, so that it would be cognitively easier for the reader to direct the reader to the conceptual framework.

Does the purple colour in the conceptual framework have any specific meaning or relationship? If so, it would be good to elaborate, as a footnote.

An average reader might not be familiar with the regions of Myanmar. It would be good to demonstrate the sampling regions and sites of them. This will also help the reader to better understand the sampling frame and mechanism, as well as the generalizability of the study to Myanmar. Just a suggestion.

Figure 3: It took me a while to understand what the left and right % values mean (26%,.... On the left and 74%,.... On the right). I understood that left is % for girls and right is the % for boys (brothers), right? Better to add a female or male symbol or a sign above the percentage to help the reader understand this figure easily.

Table 4

Suggest reformatting as for the following format.

Figure XX: Multiple Logistic Regression Predicting the Likelihood of past year parent and caregiver perpetrated physical and psychological violence

(n=514)

Variable

aOR

95% CI

P value

As of now it is a bit complex to understand with the first column title of exposure and the column titled reference group.

Also it is suggested to have a univariate analysis table prior to the multiple logistic regression table.

Consider adding comparison table for exposed and unexposed before the multiple logistic regression table.

Limitations

It is worth discussing the role of qualitative methods in improving the depth of understanding a topic like this.

Conclusions: It may be good to discuss how the Circles could be used to address this issue. Can the program reach out to parents? Can awareness raising be conducted to the parents through the circle program? I was wondering what would caregivers think about these findings? Can we use these findings to advocate for the prevention of violence by caregivers?

**Do you want your identity to be public for this peer review?** For information about this choice, including consent withdrawal, please see our Privacy Policy

Reviewer #1: No

Reviewer #2: **Yes: ** Novil Wijesekara

---

## [Decision Letter · Decision Letter 1]

9 Oct 2025

Conflict, caregiver violence and gendered parenting: A cross-sectional study among adolescent girls and young women participating in a girls’ empowerment programme in Myanmar

PGPH-D-25-00963R1

Dear Dr. Pearson,

We are pleased to inform you that your manuscript 'Conflict, caregiver violence and gendered parenting: A cross-sectional study among adolescent girls and young women participating in a girls’ empowerment programme in Myanmar' has been provisionally accepted for publication in PLOS Global Public Health.

Best regards,

Zahra Zeinali, MD MPH DrPH

Academic Editor

Reviewer Comments (if any, and for reference):

Reviewer's Responses to Questions

**Comments to the Author**

Reviewer #1: All comments have been addressed

Reviewer #2: All comments have been addressed

publication criteria?

Reviewer #1: Yes

Reviewer #2: Yes

3. Has the statistical analysis been performed appropriately and rigorously?

Reviewer #1: Yes

Reviewer #2: Yes

4. Have the authors made all data underlying the findings in their manuscript fully available (please refer to the Data Availability Statement at the start of the manuscript PDF file)?

Reviewer #1: No

Reviewer #2: No

5. Is the manuscript presented in an intelligible fashion and written in standard English?

Reviewer #1: Yes

Reviewer #2: Yes

Reviewer #1: Accept this manuscript

Reviewer #2: 4- The authors have justified why the data cannot be made publicly available, which is an acceptable reason, as per my understanding.

**Do you want your identity to be public for this peer review?** For information about this choice, including consent withdrawal, please see our Privacy Policy

Reviewer #1: No

Reviewer #2: No
